# Can kinesio tape negatively affect the treatment by creating a hard floor in plantar fasciitis treatment? A randomized clinical trial

Tuğba Kocahan[1]*, Aydan Örsçelik[1¤a], Hüseyin Günaydın[1], Gökhan Büyüklüoğlu[1], Burak Karaaslan[2], Erdoğan Asar[3], Bihter Akınoğlu[4¤b]

1 Health Sciences University, Gülhane Faculty of Medicine, Department of Sports Medicine, Ankara, Türkiye, 2 Özel Hayat Doğa Polyclinic, Ankara, Türkiye, 3 Health Sciences University, Gülhane Faculty of Medicine, Division of Basic Medical Sciences, Department of Medical Informatics, Ankara, Türkiye, 4 Ankara Yıldırım Beyazıt University, Faculty of Health Sciences, Department of Physiotherapy and Rehabilitation, Ankara, Türkiye

¤a Current Address: Health Sciences University, Gülhane Faculty of Medicine, Department of Sports Medicine, Ankara, Türkiye
¤b Current Address: Ankara Yıldırım Beyazıt University, Faculty of Health Sciences, Department of Physiotherapy and Rehabilitation, Ankara, Türkiye
* kocahantu@gmail.com

## Abstract

### Background

Radial extracorporeal shock wave therapy (r-ESWT) is accepted as one of the most effective treatment modalities for plantar fasciitis (PF). Also kinesio taping (KT) applicationis effective for the treatment of PF. However, there is not enough evidence about the effectiveness of the combination of these two modalities in the treatment of PF. This prospective randomized clinical trial was planned to investigate the acute effects of KT application in addition to r-ESWT application on pain, foot function and flexibility.

### Methods

The study was performed on 42 patients with unilateral PF, that were randomly assigned into two groups receiving four sessions of either r-ESWT or r-ESWT+KT treatment once per week. All patients performed home exercises. Patients' pain levels were evaluated via the Visual Analogue Scale (VAS), and foot function via the Foot Function Index (FFI). Flexibility was evaluated through gastro-soleus and plantar fascia flexibility tests. The evaluations were done before and one week after the final treatment. Two Way Analysis of Variance with Repeated Measures and Generalized Estimating Equations (GEE) methods were used in statistical analyses.

**Data availability statement:** All relevant data are within the paper and its Supporting Information.

**Funding:** The author(s) received no specific funding for this work.

**Competing interests:** The authors have declared that no competing interests exist.

## Results

As a result, rest pain and activity pain decreased in both groups and there is no differences between the groups (respectively p: 0,831; p: 0.331). FFI pain and disability subscores decreased and were also similar between the groups (respectively p: 0.304; p: 0.978). FFI activity limitation subscore decreased in the r-ESWT group more than r-ESWT+KT group (p: 0.002). Night pain, gastro-soleus and plantar fascia flexibility did not change in both goups (respectively p: 0.713; p: 0.413; p: 0.475).

## Conclusion

Adding KT to r-ESWT application did not create an additional pain relieving effect, on the contrary, resulted in less improvement in activity limitation. This may be related to the fact that KT application to the sole and heel region creates a hard surface on the heel. r-ESWT application may be more beneficial in PF where activity limitation are prominent.

## ClinicalTrials.gov

The registration number: NCT06516393

## Introduction

PF is a condition that affects the plantar fascia and has multiple factors in its etiology. It is the leading cause of foot pain and predominantly occurs in individuals aged 40–60 years with varying activity levels [1,2]. One-third of patients' exhibit bilateral symptoms, and the condition is more prevalent in women [3]. Chronic overload leads to inflammation and thickening of the plantar fascia. It is believed that repetitive tension during extended periods of standing or running leads to microtears at the point where the plantar fascia inserts, contributing to the development of PF. This process is chronic and degenerative rather than an acute inflammatory process [3–6]. There are lots of anatomical abnormalities as a risk factors for the development of PF [2,4–7].

Conservative treatment approaches can alleviate pain. Initially, patients may try treatments that include resting, modifying their activities, ice massages, taking analgesic medication, and using stretching techniques for a few weeks. If the heel pain persists, healthcare professionals may recommend treatments such as physical therapy modalities (e.g., ultrasound therapy, iontophoresis, laser therapy, and extracorporeal shock wave therapy), foot orthoses, night splints, and corticosteroid injections. Approximately 90–95% of patients recover with conservative treatments [2,4]. In cases of chronic, persistent PF where these methods fail, surgical intervention (plantar fasciotomy) may be considered as a final resort. It remains unclear which treatment method is most effective for PF; however, combining conservative treatments has demonstrated greater success. Insufficient evidence exists to determine the most effective approach [2,4,8].

Extracorporeal shock wave therapy (ESWT) has demonstrated efficacy in the treatment of PF due to its mechanisms of action, which include hyperstimulation analgesia, neovascularisation induction, increased blood flow to the tissue, and initiation of tissue regeneration [9–11]. A comparison of outcomes between patients treated with ESWT and those treated with other therapies has shown that the former group presents higher rates of recovery, reduction in pain scores, faster return to work, and fewer complications [10]. Therefore, ESWT is recommended as a safe and effective treatment for PF [9–11].

KT can regulate the tensile forces applied to a specific tendon or ligament to facilitate tissue repair. As a result, it has been reported that applying KT to the plantar fascia and calf muscles aids the plantar fascia's repair by mitigating the tensile force of the plantar flexors and fascia, which reduces inflammation by easing local circulation and enhances kinesthetic awareness [12,13]. It has been reported that KT treatment enhances pain reduction and quality of life in individuals with PF, at the same time, affects functionality. It has been shown that KT can improve pain, functionality and quality of life by supporting foot biomechanics in long-term applications [14].

While it is acknowledged that both r-ESWT and KT are effective in treating pain associated with PF, there is limited research regarding their combined efficacy. Given the circumstances highlighted in the literature, we hypothesize that the immediate pain-relieving effect of combining r-ESWT and KT will be superior to that of r-ESWT alone in the treatment of PF. The primary objective of this research was to examine the immediate impact of r-ESWT and KT in combination, on pain experienced during PF treatment.

## Materials and methods

### Patients

Permission to conduct the study was obtained from the Ethics Committee of Gülhane Training and Research Hospital on 13.09.2023 under the number 2023/161. The research was conducted in accordance with the Helsinki Declaration as revised in 2008. Patients who attended the Sports Medicine Outpatient Clinic of Gülhane Training and Research Hospital, were diagnosed with PF by a specialist and met the study's criteria were invited to participate. Participants who agreed to take part in the study provided written informed consent which had been approved by the ethics committee. The exact date range for participant recruitment and follow-up was 13/09/2023 - 18/04/2024.

A power analysis was conducted, taking into account the statistical methods and other relevant parameters for the study. The α (type I) error level: 0.05; the statistical power (or the β (type II) error level): 0.80, the target sample size: total 34, each group: 17, the statistical testing method: Two Way Analysis of Variance with Repeated Measures, effect size: 0.25 (medium level). The value of 0.25 was determined by expert opinion.

At the beginning of the study, a total of 50 patients were enrolled, 25 in the r-ESWT group and 25 in the r-ESWT+KT group, but eight patients withdrew from the study. After the withdrawal of eight patients from the study, the final sample size was 42 patients who met the inclusion criteria and study protocols "Fig 1". These 42 patients were diagnosed with plantar fasciitis, with a mean age of 43.7 years (range: 20–58 years), and were assigned to either the r-ESWT group (n = 18) or the r-ESWT+KT group (n = 24). After the initial evaluation, patients were randomly assigned to one of two groups by the sealed envelope method.

The inclusion criteria consisted of tenderness upon palpation of the heel, pain experienced in the plantar region for at least three months, the presence of a heel spur on a lateral radiograph of the foot, unilateral plantar fasciitis, and a willingness to participate in the study [1,4].

Exclusion criteria comprised lower extremity surgeries or traumas, systemic inflammatory disease, steroid injections administered within the past six months, pacemaker usage, coagulation disorders, use of anticoagulants, inability to perform the assessed test parameters, and unwillingness to participate in the study.

                                                                      

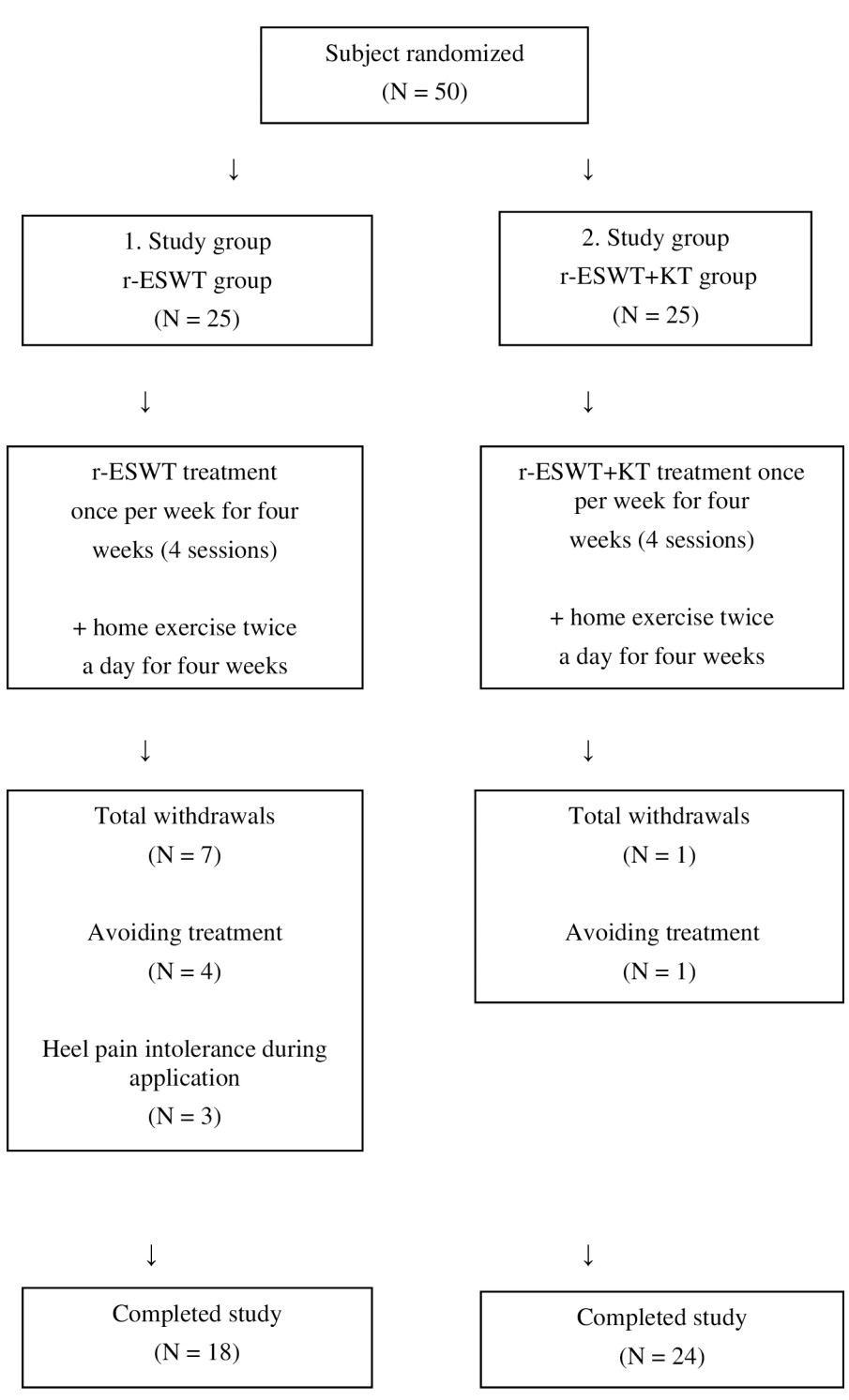

**Fig 1. Schematic presentation of the study flow.**

## Outcome measurements

All participants were assessed before the first treatment. A week following the last treatment, both groups underwent final evaluations. The demographic information form was used to record the age, height, body weight, and the dominant side of each participant. The lower limb dominance status was determined by questioning the leg that the patients naturally used to kick a ball. Pain levels of patients were evaluated using the VAS, while foot function was measured using the FFI, and flexibility was assessed through gastro-soleus and plantar fascia flexibility tests before and one week post-treatment.

## Interventions

Patients were randomly allocated into two groups following the initial assessment. All patients were given home exercise programs comprising stretching exercises for the calf muscles, Achilles tendon, and plantar fascia in addition to the treatment. The first study group received r-ESWT once a week for a total of four sessions, and the second study group received r-ESWT once a week for a total of four sessions plus KT at the end of treatment. r-ESWT and KT were performed by the same certified physiotherapist. Patients taking part in the study were instructed to solely perform the home exercise program and abstain from using orthotic support to ensure no impact on the collected data. The patient's adherence to the program, use of supports, and additional exercise were assessed during treatment sessions, and non-compliant subjects were excluded from the study.

## r-ESWT treatment

The study utilized the radial shock wave device, "DUOLITH SD-1", developed by the Storz Medical Company based in Tagerwilen, Sweden. The treatment, r-ESWT, was administered using a 15 mm diameter treatment head.

The application of r-ESWT on patients was linear, starting from the most sensitive area on the inner side of the heel and moving towards the most sensitive areas on the toes, with the patient in the prone position with ankle immobilization. No local anesthetic or analgesic medication was administered before or during the procedure. The therapy consisted of 2000 pulses, 8 Hz, and a dose of 0.3 mJ/mm$^2$, and during the therapy, ultrasound gel was employed to ensure conductivity between the head and skin [15]. A total of four r-ESWT sessions, executed once a week, were conducted "Fig 2A".

## Kinesio tape application

Participants in Study Group II received Kinesio Tex® Tape (Kinesio Holding Corporation, Albuquerque, NM, USA) application once a week for four weeks after each r-ESWT session. A single tape was applied to the patient's foot for one week (one application per week). The patient was instructed to remove the tape if it lost its adhesiveness or became loose after three or four days. The tape is flexible, waterproof, and adhesive. Its width is 5 cm, and its thickness is 0.5 mm. During the taping, the patient assumed a prone position with their knee joint flexed at a 90° angle and their ankle joint in a neutral position. The tape was sliced longitudinally into four equal parts and applied to the forefoot with 25% tension "Fig 2B" [14].

## Home exercise program

All patients were prescribed a home exercise program. The patients were asked to perform standing gastrocnemius and gastro-soleus muscle stretching exercises (standing calf stretching exercise, standing soleus muscle stretching exercise) "Figs 3 A and B", seated Achilles tendon stretching exercise (towel stretching exercise) "Fig 3C" and stepping plantar fascia stretching exercise "Fig 3D" 10 times every morning and evening for four weeks, counting up to 30 [16]. During the treatment sessions, the patient's compliance with the home exercise program was questioned by the physiotherapist and recorded.

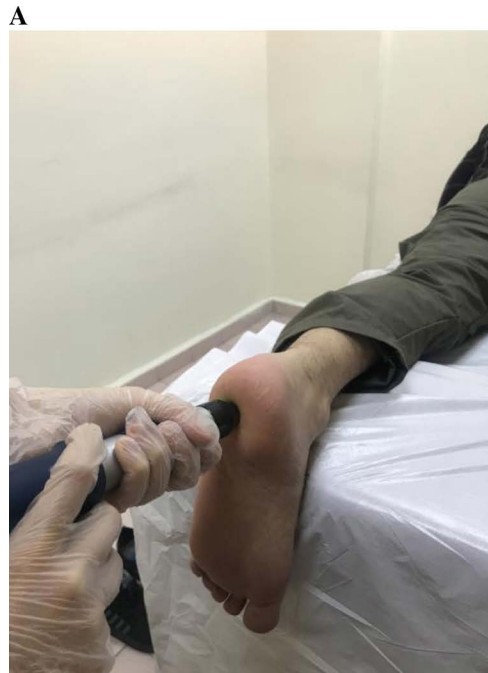
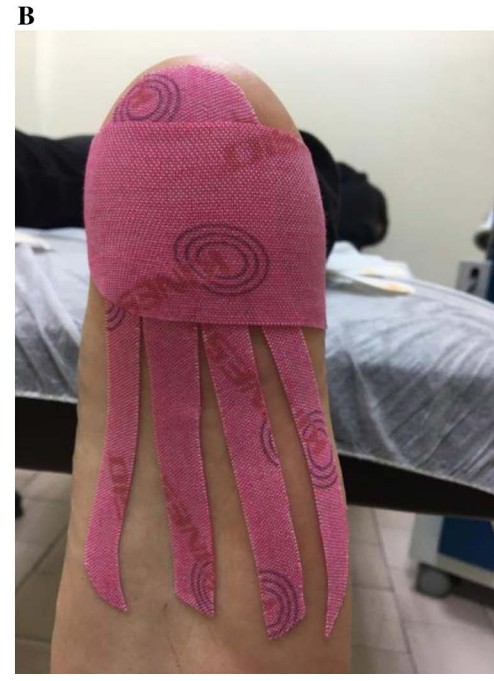

**Fig 2. Application of the treatment modalities. (A) r-ESWT application (B) KT application.**

## Evaluation tests

The pain level of the patients before and one week after the last treatment was evaluated by VAS, foot function was evaluated by FFI and flexibility was evaluated by gastro-soleus and plantar fascia flexibility tests.

**Pain Assessment.** The level of pain experienced by patients was determined using the VAS, a ruler on a 10 cm scale. The obtained values were recorded.

**Foot Function Index.** The Foot Function Index is a safe and valid foot-specific assessment tool widely used to evaluate pain, disability, and activity limitations in patients with PF. It is self-administered by the patient [17]. In the study, pain, disability, and activity limitation were objectively measured in patients and the sum was calculated. The resulting numerical data was recorded.

**Flexibility tests.** Tests were conducted with patients seated on a flat surface and their knees extended against the wall. Each test was repeated three times and the average of the three measurements was taken [18].

**Gastrocnemius-soleus flexibility test.** The participant was instructed to sit in a long sitting position with the knees extended and press the soles of their feet against the wall, then extend their arms forward and attempt to reach for their toes. The distance between the longest finger and the wall was measured in this posture [18].

**Plantar fascia flexibility measurement.** The participant sat with their legs extended in front, feet resting against a wall, and hands resting behind their back to remove hamstring tension as much as possible. Then, using dorsiflexion, the participant pulled their ankles towards themselves as much as they could. In this position, the distance between the wall and the tip of the big toe was evaluated [18].

## Statistical analysis

Statistical data analysis was performed with hypothesis tests with a significance level of 0.05. Whether the data was normally distributed or not was tested with the Shapiro-Wilk Test, and the homogeneity of variances was tested with the

A
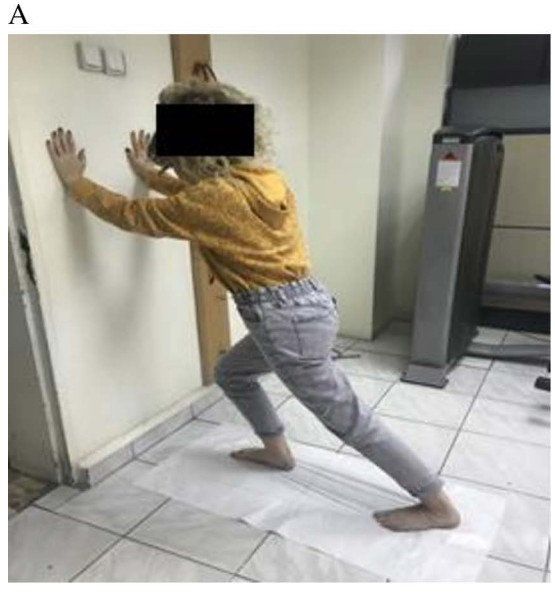

B
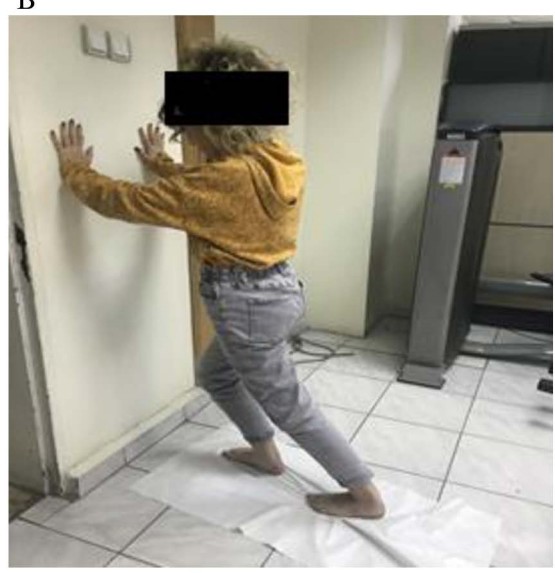

C
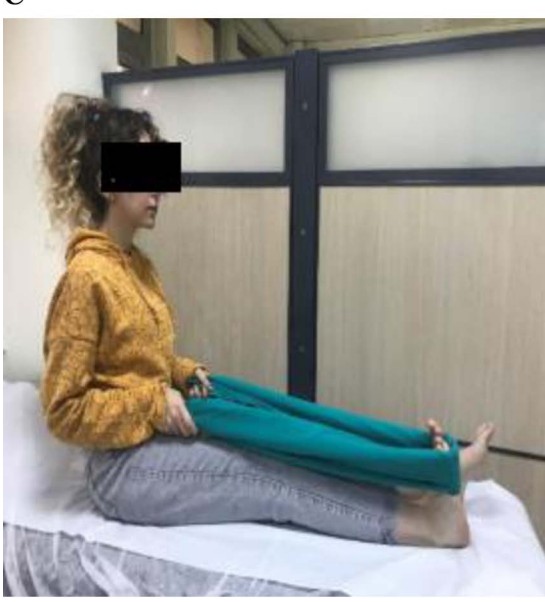

D
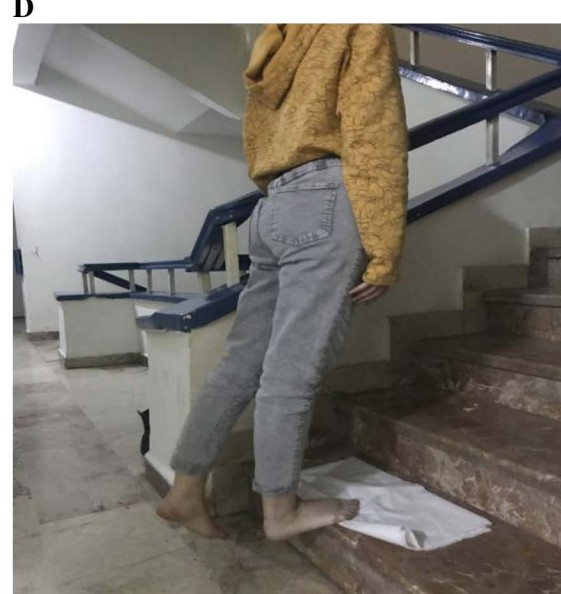

**Fig 3. Home exercises program. (A) Standing calf stretch exercise (B) Standing soleus muscle standing exercise (C) Towel stretch exercise (D) Plantar fascia stretch exercise.**

Levene Test. These analyses were performed using Jamovi (version 2.3) software. A comparisons of patients demographics were made with the Independent Samples t Test or Mann-Whitney U Test, depending on whether the parametric test assumptions were met or not.

Main effects and interaction effects were tested. Tests were analyzed using Two Way Analysis of Variance with Repeated Measures for variables normal (Gaussian) distributed. Analyses were performed using Generalized Estimating Equations (GEE) for variables not showing normal distribution. In cases where the interaction term (time*group) was

significant, the results were interpreted; results regarding main effects were not taken into account. If the interaction term was not significant, analyses were performed considering the main effects.

## Results

Table 1 displays patients' demographic characteristics. The two groups were found to be comparable in terms of age, height, body weight, and body mass index (respectively p: 0.207; p: 0.186; p:0.102; p: 0.567) "Table 1".

Descriptive statistics for normally distributed variables are presented in Table 2; descriptive statistics for non-normally distributed variables are presented in Table 3.

There was no statistically significant difference between the groups in terms of rest pain, the group variable was not significant (p: 0.831; Partial Eta Squared ($\eta^2$): 0.001); the effect of time was found to be statistically significant, the rest pain decreased in both groups post-treatment, the effect of time was found to be substantial (p: 0.002; Partial $\eta^2$: 0.217); however, there was no statistically significant difference between the groups in terms of rest pain post-treatment, the time*group interaction was not significant (p: 0.119; Partial $\eta^2$: 0.060) "Table 4".

There was no statistically significant difference between the groups in terms of activity pain, the group variable was not significant (p: 0.331; Partial $\eta^2$: 0.024); the effect of time was found to be statistically significant, the activity pain

**Table 1. Patient demographics.**

| Groups | | | | | | | | | |
|---|---|---|---|---|---|---|---|---|---|
| Variable | r-ESWT (N = 18) | | | | r-ESWT+KT (N = 24) | | | | |
| | Mean±SD | Median | Range (Max-Min) | IQR 25-75 | Mean±SD | Median | Range (Max-Min) | IQR 25-75 | p |
| Age (years) | 43.7 ± 11.2 | 46.5 | 38 (58-20) | 36.0-51.5 | 48.9 ± 6.34 | 48.0 | 22 (60-38) | 44.5-54.3 | 0.207[b] |
| Height (m) | 1.67 ± 9.69 | 1.68 | 32 (182-150) | 1.60-1.76 | 1.64 ± 7.69 | 1.62 | 29 (184-155) | 1.59-1.66 | 0.186[a] |
| Weight (kg) | 80.3 ± 11.3 | 77.0 | 35 (97-62) | 72.5-90.8 | 75.0 ± 7.34 | 72.0 | 25 (90-65) | 70.0-80.8 | 0.102[b] |
| BMI kg/m$^2$) | 28.8 ± 4.8 | 29.3 | 19.4 (42.2-22.8) | 25.6-30.1 | 28.0 ± 2.2 | 28.4 | 8.84 (33.1-24.2) | 26.3-29.0 | 0.567[b] |

r-ESWT = radial extracorporeal shock wave therapy; r-ESWT+KT = radial extracorporeal shock wave therapy+kinesiotaping; SD = Sandard deviation; IQR = interquartile range;

[a]: Independent Samples t Test;

[b]:Mann-Whitney U Test

**Table 2. Descriptive statistics for normally distributed variables.**

| Groups | | | | | | | | | | |
|---|---|---|---|---|---|---|---|---|---|---|
| Variable | | Order of measurement | r-ESWT (N = 18) | | | | r-ESWT+KT (N = 24) | | | |
| | | | Mean | Standard deviation | Min value | Max value | Mean | Standard deviation | Min value | Max value |
| Rest pain | | Pre-treatment | 3.6 | 2.8 | 0.0 | 8.0 | 2.6 | 2.8 | 0.0 | 8.0 |
| | | Post-treatment | 1.0 | 2.1 | 0.0 | 6.0 | 1.7 | 2.7 | 0.0 | 9.0 |
| Activity pain | | Pre-treatment | 7.7 | 2.4 | 1.0 | 10.0 | 7.6 | 1.7 | 4.0 | 10.0 |
| | | Post-treatment | 3.4 | 2.9 | 0.0 | 8.0 | 4.6 | 2.7 | 0.0 | 10.0 |
| FFI | Pain | Pre-treatment | 55.7 | 30.6 | 16.0 | 10.0 | 61.9 | 24.6 | 29.0 | 136.0 |
| | | Post-treatment | 31.4 | 23.8 | 2.0 | 72.0 | 39.0 | 17.4 | 14.0 | 77.0 |
| | Activity limitation | Pre-treatment | 13.1 | 6.7 | 3.0 | 27.0 | 15.5 | 7.4 | 0.0 | 26.0 |
| | | Post-treatment | 4.3 | 5.8 | 0.0 | 19.0 | 11.5 | 7.5 | 0.0 | 30.0 |
| Gastrocnemius Shortness | | Pre-treatment | 8.9 | 3.0 | 2.0 | 17.0 | 8.7 | 1.7 | 5.8 | 11.5 |
| | | Post-treatment | 8.6 | 2.3 | 3.5 | 12.0 | 9.8 | 1.7 | 6.0 | 12.2 |

**Table 3. Descriptive statistics for non-normally distributed variables.**

| Groups | | | | | | | | | | | | |
|---|---|---|---|---|---|---|---|---|---|---|---|---|
| | | r-ESWT (N = 18) | | | | | r-ESWT+KT (N = 24) | | | | | |
| Variable | Order of measurement | First quartile | Median | Third quartile | Min value | Max value | First quartile | Median | Third quartile | Min value | Max value | |
| **Night pain** | Pre-treatment | 0.0 | 0.0 | 3.6 | 0.0 | 8.2 | 0.0 | 0.0 | 5.3 | 0.0 | 8.0 | |
| | Post-treatment | 0.0 | 0.0 | 0.0 | 0.0 | 3.0 | 0.0 | 0.0 | 0.0 | 0.0 | 10.0 | |
| **FFI** **Disability** | Pre-treatment | 24.3 | 56.0 | 79.5 | 9.0 | 124.0 | 48.8 | 59.0 | 70.8 | 36.0 | 162.0 | |
| | Post-treatment | 6.5 | 18.0 | 28.0 | 0.0 | 75.0 | 29.0 | 41.0 | 60.3 | 12.0 | 90.0 | |
| **Total** | Pre-treatment | 66.3 | 120.5 | 160.5 | 38.0 | 264.0 | 107.5 | 134.5 | 151.0 | 75.0 | 312.0 | |
| | Post-treatment | 16.5 | 29.0 | 72.3 | 2.0 | 140.0 | 64.3 | 86.5 | 112.0 | 40.0 | 181.0 | |
| **Plantar fascia shortness** | Pre-treatment | 5.6 | 6.3 | 7.0 | 2.1 | 8.5 | 5.0 | 5.7 | 6.2 | 4.2 | 7.0 | |
| | Post-treatment | 5.2 | 6.0 | 7.0 | 3.5 | 9.0 | 5.8 | 6.1 | 6.7 | 5.0 | 84.6 | |

**Table 4. Comparison of pre- and post-treatment evaluation results of patients.**

| Variable* | | | Sum of Squares | Degree of freedom | Mean squares | F statistic | p value |
|---|---|---|---|---|---|---|---|
| **Rest pain** | | Group | 0.37 | 1 | 0.37 | 0.046 | 0.831 |
| | | Time | 63.60 | 1 | 63.60 | 11.091 | **0.002** |
| | | Group*Time | 14.57 | 1 | 14.57 | 2.541 | 0.119 |
| **Activity pain** | | Group | 6.48 | 1 | 6.48 | 0.967 | 0.331 |
| | | Time | 274.48 | 1 | 274.48 | 53.540 | **<0.001** |
| | | Group*Time | 8.77 | 1 | 8.77 | 1.710 | 0.198 |
| **FFI** | **Pain** | Group | 986.12 | 1 | 986.12 | 1.086 | 0.304 |
| | | Time | 11407.62 | 1 | 11407.62 | 45.188 | **<0.001** |
| | | Group*Time | 9.33 | 1 | 9.33 | 0.037 | 0.848 |
| | **Activity limitation** | Group | 484.72 | 1 | 484.72 | 6.909 | **0.012** |
| | | Time | 834.21 | 1 | 834.21 | 31.185 | **<0.001** |
| | | Group*Time | 119.45 | 1 | 119.45 | 4.465 | **0.041** |
| **Gastrocnemius shortness** | | Group | 4.74 | 1 | 4.71 | 0.683 | 0.413 |
| | | Time | 2.37 | 1 | 2.37 | 0.966 | 0.332 |
| | | Group*Time | 9.78 | 1 | 9.78 | 3.981 | 0.053 |

*Normally distributed variables; Two-Way Analysis of Variance with Repeated Measures Results

decreased in both groups post-treatment, the effect of time was found to be substantial (p < 0.001; Partial ($\eta^2$: 0.572); however, there was no statistically significant difference between the groups in terms of activity pain post-treatment, the time*group interaction was not significant (p: 0.198; Partial $\eta^2$: 0.041) "Table 4".

There was no statistically significant difference between the groups in terms of FFI pain, the group variable was not significant (p: 0.304; Partial $\eta^2$: 0.026); the effect of time was found to be statistically significant, the FFI pain decreased in both groups post-treatment, the effect of time was found to be substantial (p < 0.001; Partial $\eta^2$: 0.530); however, there was no statistically significant difference between the groups in terms of FFI pain post-treatment, the time*group interaction was not significant (p: 0.848; Partial $\eta^2$: 0.001) "Table 4".

The time*group interaction is significant for FFI activity limitation (p: 0.041; Partial $\eta^2$: 0.100). The difference between the pre-treatment and post-treatment (initial and final) measurements of the FFI activity limitation score of the r-ESWT group is statistically significant (p < 0.001; Partial $\eta^2$: 0.393). The difference between the pre-treatment and post-treatment

(initial and final) measurements of the FFI activity limitation score of the r-ESWT+KT group is statistically significant (p: 0.011; Partial $\eta^2$: 0.149). The pre-treatment (initial) measurements of the FFI activity limitation score do not differ in the two treatment methods (Groups). The post-treatment (final) measurements of the FFI activity limitation score differ in terms of the two treatment methods (Groups) (p: 0.002; Partial $\eta^2$: 0.225) "Table 4".

There was no statistically significant difference between the groups in terms of gastrocnemius shortness, the group variable was not significant (p: 0.413; Partial $\eta$2:0.017); the effect of time was not statistically significant, gastrocnemius shortness did not change in both groups post-treatment, the effect of time was not significant (p < 0.332; Partial $\eta$2: 0.024); there was no statistically significant difference between the groups in terms of gastrocnemius shortness post-treatment, the time*group interaction was not significant (p: 0.053; Partial $\eta$2: 0.091) "Table 4".

There was no statistically significant difference between the groups in terms of night pain, the group variable was not significant (p: 0.713); the effect of time was not statistically significant, and the change in night pain did not differ over time (p: 0.071); there was no statistically significant difference between the groups in terms of night pain post-treatment, the time*group interaction was not significant (p: 0.336) "Table 5".

There was no statistically significant difference between the groups in terms of FFI-disability, the group variable was not significant (p: 0.978); the effect of time was found to be statistically significant, the change in FFI-disability varies over time (p: 0.002); there was no statistically significant difference between the groups in terms of FFI-disability post-treatment, the time*group interaction was not significant (p: 0.192) "Table 5".

There was no statistically significant difference between the groups in terms of FFI-total, the group variable was not significant (p: 0.639); the effect of time was found to be statistically significant, and the change in FFI-total differs over time (p: 0.001); the change in FFI-total over time did not differ according to the groups, the time*group interaction was not significant (p: 0.113) "Table 5".

There was no statistical difference between the groups in terms of plantar fascia shortness, the group variable was not significant (p: 0.475); the effect of time was not statistically significant, the change in plantar fascia shortness did not differ over time (p: 0.534); time*group interaction was not significant (p: 0.308) "Table 5".

## Discussion

This study aimed to investigate the acute impact of KT application applied with r-ESWT on pain, foot function, and flexibility in PF treatment. After treatment, rest pain and activity pain decreased in both groups and there is no differences

Table 5. Comparison of pre- and post-treatment evaluation results of patients.

| Variable* | | | Regression Coefficients | Standard Error | z statistic | p value |
|---|---|---|---|---|---|---|
| Night pain | | Group | -0.552 | 1.499 | -0.37 | 0.713 |
| | | Time | -2.593 | 1.436 | -1.81 | 0.071 |
| | | Group*Time | 0.838 | 0.872 | 0.96 | 0.336 |
| FFI | Disability | Group | -0.389 | 13.831 | -0.03 | 0.978 |
| | | Time | -40.667 | 13.058 | -3.11 | **0.002** |
| | | Group*Time | 10.333 | 7.926 | 1.3 | 0.192 |
| | Total | Group | -14.611 | 31.133 | -0.47 | 0.639 |
| | | Time | -106.792 | 30.775 | -3.47 | **0.001** |
| | | Group*Time | 29.625 | 18.68 | 1.59 | 0.113 |
| Plantar fascia shortness | | Group | -4.24 | 5.938 | -0.71 | 0.475 |
| | | Time | -3.86 | 6.199 | -0.62 | 0.534 |
| | | Group*Time | 3.832 | 3.763 | 1.02 | 0.308 |

*Non-normally distributed variables; Generalized Estimating Equations (GEE) Analysis Results

between the groups. FFI pain and disability subscores decreased and were also similar between the groups. FFI activity limitation subscore decreased in the r-ESWT group more than r-ESWT+KT group. Night pain, gastro-soleus and plantar fascia flexibility did not change in both goups.

PF is a painful condition that leads to fibrosis and degeneration of the plantar fascia, impairing daily activities for patients [6,19]. ESWT treatment, a non-invasive procedure that breaks down fibrosis and stimulates neovascularisation, aids in healing degenerative tissue in individuals suffering from persistent chronic PF [11]. Numerous studies have demonstrated the efficacy of ESWT in treating chronic plantar fasciitis, resulting in pain reduction and improved foot function over the short, medium, and long term [8,15,20–25]. Our study employed r-ESWT treatment in both participant groups, resulting in reduced pain levels in all patients and corroborating previous research findings. No additional pain reduction was observed in the group receiving KT with r-ESWT.

In contrast to our study results, Tezel et al. randomly divided 84 PF patients into two groups and applied only KT treatment to one group and only r-ESWT treatment to the other group for 6 weeks. Both KT and r-ESWT treatments improved pain levels and quality of life in PF patients, and it was reported that KT increased functionality further [14]. In our study, 4 weeks of application may have been insufficient to reveal the effectiveness of KT application in particular. However, in the study conducted by Tezel et al., the dose of r-ESWT application was 0.2 mj/mm$^2$. This application dose is different from our study. We think that the differences in results between the studies are due to the r-ESWT dose and that a dose of 0.3 mj/mm$^2$ may be more effective in the treatment of PF. Özdemir and colleagues conducted a randomized trial with 45 patients diagnosed with plantar fasciitis. Group 1 received five sessions of ESWT+low-dye KT, Group 2 received ESWT+Sham-taping, and Group 3 received only ESWT. The patients were divided into three groups. The patients were assessed using VAS, heel tenderness index, and FFI at the start and end of the treatment, as well as at the 4-week follow-up. Although the addition of low-dye Kinesio taping to ESWT was more effective than the other two treatment groups in improving foot function, no significant benefit on PF-related pain and heel tenderness was reported [20]. Similarly, this study found that the use of KT in addition to r-ESWT did not provide any additional effect on pain. However, it was determined that only the application of r-ESWT increased foot functionality. It is possible to explain this result by analyzing the FFI parameters. In the r-ESWT+KT group, the requirement for patients to move with a band on the soles of their feet during standing and walking, and the resulting anxiety about the band slipping off or bunching up under the foot and creating an uneven surface, may have had a negative impact. It may also have created a difference in leg length by creating height at the sole of the foot. In addition, in the KT application, the overlapping of two layers of tape and the application of these tapes with 25% tension may have created a hard base on the already painful heel. Just like a patient with PF is more comfortable in soft-soled shoes but feels more pain when walking barefoot on the ground. Studies in the literature show that exercises involving areas such as the elbow, wrist, and back, which do not bear weight and where the band is unlikely to dislodge during function, are commonly practiced [26–29]. Nonetheless, the omission of inquiries about this could be viewed as a limitation of our study. Further research on this subject is warranted.

There are contradictory results related to the effect of KT applications in many studies in the literatüre with the hypothesizes that KT application provides fascial mobility by lifting and enhancing circulation on the fascia [12,13].No study investigating the effect of r-ESWT and KT application on gastro-soleus and plantar fascia flexibility was found in the literature. Nevertheless, Castro-Mendez et al. randomly assigned 57 PF patients to two groups and applied Dynamic Tape® to one group and low-dye taping to the other group on the gastrocnemius-achilles-plantar system for one week to investigate the effectiveness of these techniques on ankle range of motion. At the conclusion of the study, it was reported that there was no alteration in the ankle range of motion [30]. Also our findings suggest that r-ESWT+KT application also the r-ESWT application did not enhances gastro-soleus and plantar fascia flexibility.

The study's limitations include the absence of a control group with exercise and/or no treatment, and the lack of analysis on the long-term effects of the study. The diagnosis of PF was made through radiography, which can be considered a strength. However, the study's uniqueness and strength lie in demonstrating that the plantar fascia technique, a commonly

employed KT application for musculoskeletal injuries, does not yield any additional effect on pain or foot functionality in patients with PF. In fact, it may even cause a decrease in functionality growth.

## Conclusion

Following the study investigating the acute impact of r-ESWT and adding of KT to r-ESWT on pain, functionality, and flexibility in treating PF, it was concluded that adding of KT to r-ESWT did not provide an additional effect on pain and foot function due to PF in the acute phase. Adding KT to r-ESWT application did not create an additional pain relieving effect, on the contrary, resulted in less improvement in activity limitation. This may be related to the fact that KT application to the sole and heel region creates a hard surface on the heel. r-ESWT application may be more beneficial in PF where activity limitation are prominent. The findings may aid in PF treatment management.

## Supporting Information

**S1 Checklist. CONSORT 2010 checklist of information to include when reporting a randomised trial.**
(DOC)

**S1 Study Protocol. Clinical Research Ethics Committee Document.**
(DOCX)

**S2 Study Protocol. Clinical Research Ethics Committee Document-translate.**
(DOCX)

**S1 Dataset. Dataset of analysis using Two-Way Analysis of Variance.**
(XLSX)

**S2 Dataset. Dataset of Generalized Estimating Equations.**
(XLSX)

**S3 Dataset. Main data.**
(XLSX)

## Author contributions

**Conceptualization:** Tuğba Kocahan, Aydan Örsçelik, Bihter Akınoğlu.

**Data curation:** Tuğba Kocahan, Aydan Örsçelik, Hüseyin Günaydın, Gökhan Büyüklüoğlu, Burak Karaaslan, Bihter Akınoğlu.

**Formal analysis:** Erdoğan Asar.

**Funding acquisition:** Tuğba Kocahan.

**Investigation:** Tuğba Kocahan, Bihter Akınoğlu.

**Methodology:** Erdoğan Asar, Bihter Akınoğlu.

**Project administration:** Tuğba Kocahan, Bihter Akınoğlu.

**Resources:** Tuğba Kocahan, Aydan Örsçelik, Hüseyin Günaydın, Gökhan Büyüklüoğlu, Burak Karaaslan.

**Software:** Erdoğan Asar.

**Supervision:** Tuğba Kocahan, Bihter Akınoğlu.

**Validation:** Tuğba Kocahan, Erdoğan Asar.

**Visualization:** Tuğba Kocahan, Erdoğan Asar.

**Writing – original draft:** Tuğba Kocahan, Bihter Akınoğlu.

**Writing – review & editing:** Tuğba Kocahan, Gökhan Büyüklüoğlu, Bihter Akınoğlu.

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
