## [Decision Letter · Decision Letter 0]

2 Sep 2024

PONE-D-24-26813Can kinesio tape negatively affect the treatment by creating a hard floor in plantar fasciitis treatment? A randomized clinical trialPLOS ONE

Dear Dr. Kocahan,

Thank you for submitting your manuscript to PLOS ONE. After careful consideration, we feel that it has merit but does not fully meet PLOS ONE’s publication criteria as it currently stands. Therefore, we invite you to submit a revised version of the manuscript that addresses the points raised during the review process.

We look forward to receiving your revised manuscript.

Kind regards,

Domiziano Tarantino, MD

Academic Editor

PLOS ONE

Journal Requirements: When submitting your revision, we need you to address these additional requirements. 1. Please ensure that your manuscript meets PLOS ONE's style requirements, including those for file naming. The PLOS ONE style templates can be found at https://journals.plos.org/plosone/s/file?id=wjVg/PLOSOne_formatting_sample_main_body.pdf and https://journals.plos.org/plosone/s/file?id=ba62/PLOSOne_formatting_sample_title_authors_affiliations.pdf 2. In the online submission form, you indicated that "The data underlying the results presented in the study are available from Health Sciences University, Gülhane Faculty of Medicine, Department of Sports Medicine, Ankara, Türkiye, by asking the correspounding author, Tuğba KOCAHAN." All PLOS journals now require all data underlying the findings described in their manuscript to be freely available to other researchers, either 1. In a public repository, 2. Within the manuscript itself, or 3. Uploaded as supplementary information.This policy applies to all data except where public deposition would breach compliance with the protocol approved by your research ethics board. If your data cannot be made publicly available for ethical or legal reasons (e.g., public availability would compromise patient privacy), please explain your reasons on resubmission and your exemption request will be escalated for approval. 3. Please include captions for your Supporting Information files at the end of your manuscript, and update any in-text citations to match accordingly. Please see our Supporting Information guidelines for more information: http://journals.plos.org/plosone/s/supporting-information.

Reviewers' comments:

Reviewer's Responses to Questions

**Comments to the Author**

1. Is the manuscript technically sound, and do the data support the conclusions?

Reviewer #1: Yes

Reviewer #2: No

Reviewer #3: No

2. Has the statistical analysis been performed appropriately and rigorously? 

Reviewer #1: I Don't Know

Reviewer #2: No

Reviewer #3: No

3. Have the authors made all data underlying the findings in their manuscript fully available?

Reviewer #1: Yes

Reviewer #2: No

Reviewer #3: No

4. Is the manuscript presented in an intelligible fashion and written in standard English?

Reviewer #1: Yes

Reviewer #2: Yes

Reviewer #3: Yes

5. Review Comments to the Author

Reviewer #1: The study from its title is presuming to give a neutral or a known result but it still emphasized if KT and R-ESWT will help in acute care of PF. Systematic reviews and meta- analysis mentioned in the references of the study also suggests that KT or r-ESWT can be aa optional treatment.

Good effort.

Reviewer #2: A two arm randomized clinical study was conducted which aimed to compare pain scores, Foot Function Index (FFI), activity limitation, gastroc-soleus and plantar fascia flexibility test scores in patients with plantar fasciitis. The conclusions are unclear due to multiple comparison issues.

Major revisions:

To avoid issues with multiple comparisons, provide more comprehensive analyses by testing the interaction effect of group by time. If the interaction effect is significant, provide an interpretation of the results, but do not test main effects because the tests for main effects are uninteresting in light of significant interactions. If the interaction effects are non-significant, drop the interaction effects from the model and test the main effects. Determining which results to present when testing interactions is often a multi-step process. A comprehensive reanalysis is required.

Minor revisions:

1- In the abstract, briefly identify the statistical methods used to estimate the p-values.

2- Line 123: The power calculation should include: (1) the estimated outcomes in each group; (2) the α (type I) error level; (3) the statistical power (or the β (type II) error level); (4) the target sample size and (5) the statistical testing method and (6) for continuous outcomes, the standard deviation of the measurements. Consider replacing “ideal” with “target” in line 124.

3- Tables: The columns for the “Test Statistic” can be removed from the tables.

4- If data is normally distributed, summarize using mean and standard deviation. When the distribution is non-normal, summarize with median, first and third quartiles.

Reviewer #3: While the subject is interesting and potentially insightful for treating plantar fasciitis, the manuscript contains several issues that need to be addressed.

The background is vague and insufficient. Although the abstract is subject to word limitations, the authors have not provided a clear rationale for their work.

The phrase "we believe" is not typically appropriate for use in abstract conclusions and should be revised.

The term "hard floor" is unclear. Kinesio Tape (KT), as an elastic tape, differs from rigid tape in that it does not limit the range of motion, especially with the 25% tension applied, as mentioned in the methods section. More importantly, the term "hard floor," which appears in the title and abstract, is neither explained nor discussed in the text. This needs clarification.

Lines 57-93 are overly lengthy for the background on plantar fasciitis. These sections should be summarized, retaining only the elements directly relevant to the study.

The introduction mentions various factors that can affect plantar fasciitis and its treatment. The methods section should clarify whether all these factors were considered and controlled in the study.

The authors refer to "subgrouping the subjects and then randomizing them into one of two groups," which suggests stratified block randomization. This method ensures an equal number of subjects in each group, and the text should explicitly clarify this.

Furthermore, several confounding factors were noted in the introduction. It is crucial to specify which of these were controlled in the study groups.

The inclusion criteria lack proper references and should be cited. Similarly, there are no references for the parameters or methods used for r-ESWT, and the flexibility tests also require appropriate citations.

It is unusual to leave KT on for one week without changing it. The mechanical properties of KT diminish over time, particularly on a weight-bearing area such as the foot, which is often enclosed in a shoe with socks. Leaving the tape unchanged for a week raises doubts about the validity of the study's results.

The authors need to explicitly mention which variables were normally distributed and specify the tests used for analysis. If the data were normally distributed, a mixed ANOVA (with both within- and between-subject factors) would be the appropriate statistical method.

Since the statistical methods are not clearly explained, the results cannot be thoroughly reviewed. Additionally, effect sizes should be reported alongside p-values.

There is a noticeable bias toward r-ESWT throughout the manuscript. Given that there is no significant difference between the two groups for most variables, neither treatment can be considered superior. The conclusion suggesting that KT with r-ESWT could be inferior, as mentioned in lines 276-280, seems biased and is not supported by the data. The authors should discuss the results objectively without bias.

Finally, Reference 14 is very similar to the authors' work and should be discussed more thoroughly in both the introduction and discussion sections.

6. PLOS authors have the option to publish the peer review history of their article (what does this mean? ). If published, this will include your full peer review and any attached files.

**Do you want your identity to be public for this peer review?** For information about this choice, including consent withdrawal, please see our Privacy Policy .

Reviewer #1: **Yes: ** PREM KUMAR BHOJARA

Reviewer #2: No

Reviewer #3: **Yes: ** Sahar Boozari

---

## [Author Response · Author response to Decision Letter 1]

1 Nov 2024

Dear Editor,

The Data-sets were uploaded as three files under the supporting files heading.

The necessary revisions have been made to the manuscript.

The revised manuscript has been uploaded to the system.

---

## [Decision Letter · Decision Letter 1]

15 Dec 2024

PONE-D-24-26813R1Can kinesio tape negatively affect the treatment by creating a hard floor in plantar fasciitis treatment? A randomized clinical trialPLOS ONE

Dear Dr. Kocahan,

Thank you for submitting your manuscript to PLOS ONE. After careful consideration, we feel that it has merit but does not fully meet PLOS ONE’s publication criteria as it currently stands. Therefore, we invite you to submit a revised version of the manuscript that addresses the points raised during the review process.

We look forward to receiving your revised manuscript.

Kind regards,

Domiziano Tarantino, MD

Academic Editor

PLOS ONE

Journal Requirements:

Reviewers' comments:

Reviewer's Responses to Questions

**Comments to the Author**

1. If the authors have adequately addressed your comments raised in a previous round of review and you feel that this manuscript is now acceptable for publication, you may indicate that here to bypass the “Comments to the Author” section, enter your conflict of interest statement in the “Confidential to Editor” section, and submit your "Accept" recommendation.

Reviewer #2: (No Response)

Reviewer #3: All comments have been addressed

2. Is the manuscript technically sound, and do the data support the conclusions?

Reviewer #2: Yes

Reviewer #3: Yes

3. Has the statistical analysis been performed appropriately and rigorously? 

Reviewer #2: Yes

Reviewer #3: Yes

4. Have the authors made all data underlying the findings in their manuscript fully available?

Reviewer #2: Yes

Reviewer #3: No

5. Is the manuscript presented in an intelligible fashion and written in standard English?

Reviewer #2: Yes

Reviewer #3: Yes

6. Review Comments to the Author

Reviewer #2: Minor revisions:

1- Line 55: Typographical error: Replace "chance" with "change".

2- Provide more precise p-values rather than p< 0.05 or p>0.05.

Reviewer #3: (No Response)

7. PLOS authors have the option to publish the peer review history of their article (what does this mean? ). If published, this will include your full peer review and any attached files.

**Do you want your identity to be public for this peer review?** For information about this choice, including consent withdrawal, please see our Privacy Policy .

Reviewer #2: No

Reviewer #3: **Yes: ** Sahar Boozari

---

## [Author Response · Author response to Decision Letter 2]

16 Dec 2024

Dear Editor,

Thank you for your comments and attention.

The correction you requested has been made and precise p-values are provided.

Excel files belonging to the study have been shared. Due to our Hospital's ethical policies patient names remain anonymous.

Best regards.

---

## [Decision Letter · Decision Letter 2]

13 Jan 2025

PONE-D-24-26813R2Can kinesio tape negatively affect the treatment by creating a hard floor in plantar fasciitis treatment? A randomized clinical trialPLOS ONE

Dear Dr. Kocahan,

Thank you for submitting your manuscript to PLOS ONE. After careful consideration, we feel that it has merit but does not fully meet PLOS ONE’s publication criteria as it currently stands. Therefore, we invite you to submit a revised version of the manuscript that addresses the points raised during the review process.

We look forward to receiving your revised manuscript.

Kind regards,

Domiziano Tarantino, MD

Academic Editor

PLOS ONE

Journal Requirements:

Reviewers' comments:

Reviewer's Responses to Questions

**Comments to the Author**

1. If the authors have adequately addressed your comments raised in a previous round of review and you feel that this manuscript is now acceptable for publication, you may indicate that here to bypass the “Comments to the Author” section, enter your conflict of interest statement in the “Confidential to Editor” section, and submit your "Accept" recommendation.

Reviewer #2: (No Response)

2. Is the manuscript technically sound, and do the data support the conclusions?

Reviewer #2: Yes

3. Has the statistical analysis been performed appropriately and rigorously? 

Reviewer #2: Yes

4. Have the authors made all data underlying the findings in their manuscript fully available?

Reviewer #2: Yes

5. Is the manuscript presented in an intelligible fashion and written in standard English?

Reviewer #2: Yes

6. Review Comments to the Author

Reviewer #2: Minor revision:

Line 298: Typographical error: Replace "chance" with "change".

7. PLOS authors have the option to publish the peer review history of their article (what does this mean? ). If published, this will include your full peer review and any attached files.

**Do you want your identity to be public for this peer review?** For information about this choice, including consent withdrawal, please see our Privacy Policy .

Reviewer #2: No

---

## [Author Response · Author response to Decision Letter 3]

13 Jan 2025

Dear Editor,

The necessary revisions have been made to the manuscript.

The revised manuscript has been uploaded to the system.

Kind regards

---

## [Decision Letter · Decision Letter 3]

21 Mar 2025

Can kinesio tape negatively affect the treatment by creating a hard floor in plantar fasciitis treatment? A randomized clinical trial

PONE-D-24-26813R3

Dear Dr. Kocahan,

We’re pleased to inform you that your manuscript has been judged scientifically suitable for publication and will be formally accepted for publication once it meets all outstanding technical requirements.

Kind regards,

Domiziano Tarantino, MD

Academic Editor

PLOS ONE

Additional Editor Comments (optional):

Reviewers' comments:

Reviewer's Responses to Questions

**Comments to the Author**

1. If the authors have adequately addressed your comments raised in a previous round of review and you feel that this manuscript is now acceptable for publication, you may indicate that here to bypass the “Comments to the Author” section, enter your conflict of interest statement in the “Confidential to Editor” section, and submit your "Accept" recommendation.

Reviewer #2: All comments have been addressed

2. Is the manuscript technically sound, and do the data support the conclusions?

Reviewer #2: (No Response)

3. Has the statistical analysis been performed appropriately and rigorously? 

Reviewer #2: (No Response)

4. Have the authors made all data underlying the findings in their manuscript fully available?

Reviewer #2: (No Response)

5. Is the manuscript presented in an intelligible fashion and written in standard English?

Reviewer #2: (No Response)

6. Review Comments to the Author

Reviewer #2: (No Response)

7. PLOS authors have the option to publish the peer review history of their article (what does this mean? ). If published, this will include your full peer review and any attached files.

**Do you want your identity to be public for this peer review?** For information about this choice, including consent withdrawal, please see our Privacy Policy .

Reviewer #2: No

---

## [Editor Report · Acceptance letter]

PONE-D-24-26813R3

PLOS ONE

Dear Dr. Kocahan,

I'm pleased to inform you that your manuscript has been deemed suitable for publication in PLOS ONE. Congratulations! Your manuscript is now being handed over to our production team.

Kind regards,

on behalf of

Dr. Domiziano Tarantino

Academic Editor

PLOS ONE